# A Dual-Stream Neural Network Explains the Functional Segregation of Dorsal and Ventral Visual Pathways in Human Brains

**Minkyu Choi[1], Kuan Han[1],**
**Xiaokai Wang[2], Yizhen Zhang[1,3], and Zhongming Liu[1,2]**

[1] Department of Electrical Engineering and Computer Science, University of Michigan, Ann Arbor, MI 48109
[2] Department of Biomedical Engineering, University of Michigan, Ann Arbor, MI 48109
[3] Department of Neurological Surgery , University of California, San Francisco, San Francisco, CA 94143
`{cminkyu, kuanhan, xiaokaiw, zhyz, zmliu}@umich.edu`

## Abstract

The human visual system uses two parallel pathways for spatial processing and object recognition. In contrast, computer vision systems tend to use a single feed-forward pathway, rendering them less robust, adaptive, or efficient than human vision. To bridge this gap, we developed a dual-stream vision model inspired by the human eyes and brain. At the input level, the model samples two complementary visual patterns to mimic how the human eyes use magnocellular and parvocellular retinal ganglion cells to separate retinal inputs to the brain. At the backend, the model processes the separate input patterns through two branches of convolutional neural networks (CNN) to mimic how the human brain uses the dorsal and ventral cortical pathways for parallel visual processing. The first branch (WhereCNN) samples a global view to learn spatial attention and control eye movements. The second branch (WhatCNN) samples a local view to represent the object around the fixation. Over time, the two branches interact recurrently to build a scene representation from moving fixations. We compared this model with the human brains processing the same movie and evaluated their functional alignment by linear transformation. The WhereCNN and WhatCNN branches were found to differentially match the dorsal and ventral pathways of the visual cortex, respectively, primarily due to their different learning objectives, rather than their distinctions in retinal sampling or sensitivity to attention-driven eye movements. These model-based results lead us to speculate that the distinct responses and representations of the ventral and dorsal streams are more influenced by their distinct goals in visual attention and object recognition than by their specific bias or selectivity in retinal inputs. This dual-stream model takes a further step in brain-inspired computer vision, enabling parallel neural networks to actively explore and understand the visual surroundings.

## 1   Introduction

The human visual system comprises two parallel and segregated streams of neural networks: the "where" stream and the "what" stream [1]. The "where" stream originates from magnocellular retinal ganglion cells and extends along the dorsal visual cortex. The "what" stream originates from parvocellular retinal ganglion cells and extends along the ventral visual cortex [2]. The two streams exhibit selective responses to different aspects of visual stimuli [3]. The "where" stream is tuned to coarse but fast information from a wide view, while the "what" stream is selective to fine but slow

37th Conference on Neural Information Processing Systems (NeurIPS 2023).

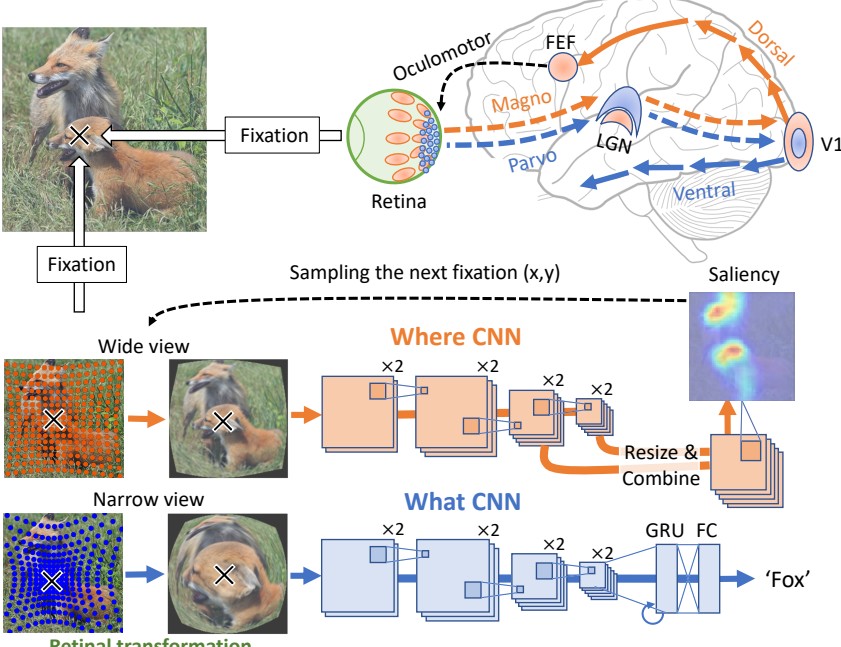

Figure 1: **Brain-inspired dual-stream vision model.** The top illustrates the subcortical (dashed arrows) and cortical (solid arrows) pathways for parallel visual processing in the brain. Given a scene (e.g., "two foxes on the lawn"), the retina samples incoming light relative to the fixation of the eyes (shown as the cross). Magnocellular (orange) and parvocellular (blue) retinal ganglion cells encode complementary visual information into two sets of retinal inputs relayed onto separate layers in the lateral geniculate nuclei (LGN) and further onto different neurons in the primary visual cortex (V1). Within V1, the relative ratio of magnocellular vs. parvocellular projections is higher for the periphery and lower for the fovea. Beyond V1, the magnocellular pathway continues along the dorsal visual cortex towards the intraparietal areas and further onto the frontal eye field (FEF) for oculomotor control, while the parvocellular pathway continues along the ventral visual cortex towards the inferior temporal cortex and further onto the superior temporal areas for semantic cognition. The bottom illustrates our model architecture including WhereCNN and WhatCNN. The model's frontend mimics the human retina and generates two separate input patterns relative to the fixation. One pattern is wider but coarser while the other is narrower but finer, providing the respective inputs to WhereCNN and WhatCNN. With the wide-view input, WhereCNN generates a probability map of saliency from which the next fixation is sampled. With a narrow-view input, WhatCNN generates an object representation per each fixation and constructs a scene representation recurrently from multiple fixations.

information from a narrow view [2, 4]. The two streams are thought to serve different purposes. The "where" stream zooms out for spatial analysis [5], visual attention [6, 7, 8], and guiding actions [9] such as eye movements [10], while the "what" stream zooms in to recognize the object around the fixation [11]. While being largely parallel, the two streams interact with each other [12]. In one way of their interaction, the "where" stream decides where to look next and guides the "what" stream to focus on a salient location for visual perception. As the eyes move around the visual environment, the interaction between the "where" and "what" streams builds a scene representation by accumulating object representations over time and space. This dual-stream architecture allows the brain to efficiently process visual information and support dynamic visual behaviors [13].

In contrast, computer vision systems tend to use a single stream of feedforward processing, acting as passive observers that sample visual information all at once with fixed and uniform patterns [14, 15, 16]. Compared to human vision, this processing is less robust, especially given adversarial attacks [17, 18]; it is less efficient since it samples visual information equally regardless of salience or nuisance [19]; it is less adaptive, lacking spatial attention for active sensing [20, 21]. These distinctions define a major gap between human and computer vision. Many visual tasks that are

straightforward for humans are still challenging for machines [22, 23]. Therefore, computer vision may benefit from taking further inspiration from the brain by using a dual-stream architecture to learn adaptive and robust visual behaviors.

To gain insights into the computational mechanisms of human vision, researchers have developed image-computable models by utilizing goal-driven deep neural networks that simulate human perceptual behavior. In particular, convolutional neural networks (CNNs) are leading models of visual perception, capturing the hierarchical processing by the brain's ventral visual stream [24, 25, 26, 27, 28]. Previous models of this nature commonly utilize CNNs trained through supervised learning [24, 27, 25, 29, 26, 30], adversarial training [31, 32], unsupervised learning [33, 34], or self-supervised learning [35, 36, 37]. However, models of the dorsal stream remain relatively under-explored, despite few studies [38, 39, 40, 41]. Existing testing of these models has primarily focused on static images presented briefly to the fovea, thus limiting their assessment to a narrow range of visual behaviors and processes [42]. A more comprehensive approach is needed to develop models that incorporate both dorsal and ventral stream processing and to assess those models against brain responses when humans engage both the dorsal and ventral streams to freely explore complex and dynamic visual environments, which may be simulated in experimental settings [43].

To meet this need, we have developed a dual-stream model to mimic the parallel ventral and dorsal streams in the human brain [1, 2, 3, 9]. The model includes two branches of convolutional neural networks: WhereCNN and WhatCNN, which share the same architecture but receive distinct visual inputs and generate different outputs. WhereCNN samples a wide view to learn spatial attention and where to direct the subsequent gaze, while WhatCNN samples a narrow view to learn object representations. By taking multiple gazes at a given scene, the model sequentially samples the salient locations and progressively constructs a scene representation over both space and time. To evaluate this dual-stream model as a model of the human visual system, we have tested its ability to reproduce human gaze behavior and predict functional brain scans from humans watching a movie with unconstrained eye movements. Our hypothesis is that the model's WhereCNN and WhatCNN branches can effectively predict the brain responses along the brain's dorsal and ventral visual pathways, respectively. In addition, we have also conducted experiments to evaluate the underlying factors contributing to the functional segregation of the brain's dorsal and ventral visual streams. Of particular interest were the relative contributions of retinal sampling, spatial attention, and attention-guided eye movement in shaping the function of the dorsal stream and its interplay with the ventral stream during dynamic natural vision.

## 2 Related Works

### 2.1 Dorsal-stream vision

Image-computable models of the brain's dorsal stream have been relatively limited compared to models of the ventral stream. Previous work has attempted to model the dorsal stream by training deep neural networks to detect motion [38] or classify actions [40] using video inputs. However, these models do not fully capture the neuroscientific understanding that the dorsal stream is involved in locating objects and guiding actions, leading to its designation as the "where" or "how" visual pathway. More recent work by Mineault et al. focused on training a dorsal-stream model to emulate human head movements during visual exploration [39]. Additionally, Bakhtiari et al. utilized predictive learning to train parallel pathways and observed the ventral-like and dorsal-like representations as an emergent consequence of structural segregation [41]. However, no prior work has explored neural network models that emulate how the dorsal stream learns spatial attention and guides eye movements for visual navigation.

### 2.2 Spatial attention and eye movement

Prior research in the field of computer vision has attempted to train models to attend to and selectively focus on salient objects within a scene [21, 44, 45], rather than processing the entire scene as a whole. This approach aligns with the brain's mechanism of spatial attention, where the dorsal stream acts as a global navigator, and the ventral stream functions as a local perceiver. In line with this mechanism, previous studies have employed dual-stream neural networks that process global and local features in parallel, aiming to achieve enhanced computational efficiency as a unified system [46, 44, 47, 48, 49].

However, these models do not fully replicate the way human eyes sample visual inputs during active exploration of the scene and thus still fall short in biological relevance.

### 2.3 Foveated vision and retinal transformation

The human retina functions as a sophisticated camera that intelligently samples and transmits visual information. It exhibits the highest visual acuity in the central region of the visual field, a phenomenon referred to as foveated vision [50, 51, 52]. In contrast, peripheral vision uses lower spatial acuity but higher temporal sensitivity, making it better suited for detecting motion. These properties of retinal sampling are potentially useful for training neural networks to enhance performance [53, 54, 55, 56] or robustness [57, 58, 19], augment data or synthesize images [59]. The retina also transmits information to the brain using distinct types of cells. The magnocellular and parvocellular retinal ganglion cells have different distributions, selectivity, and relay information through largely separate pathways. Taken together, the retina transforms visual information into segregated inputs for parallel visual processing. This biological mechanism has not been systematically investigated.

Unlike the above prior works, our work combines multiple biologically inspired mechanisms into an integral learnable model. It uses a frontend inspired by the human retina and applies complementary retinal sampling and transformation. It uses two parallel pathways inspired by the human dorsal and ventral streams. It uses spatial attention and object recognition as the distinct learning objectives for training the two pathways. It further uses the attention to move fixations and thus allows the two pathways to interact for active sensing. Although these mechanisms have been explored separately in prior studies, their combination is novel and motivates our work to build and test such a model against the human brain and behavior in a naturalistic condition of freely watching a movie.

## 3 Methods

In our model, WhereCNN and WhatCNN serve distinct functions in processing objects within a scene. WhereCNN identifies the spatial location of an object, determining "where" it is situated, while WhatCNN focuses on recognizing the identity of the object, determining "what" it is. When multiple objects are present in a scene, WhereCNN learns spatial attention and "how" to sequentially locate and fixate on each object. This allows the model to selectively attend to different objects in a sequence, mirroring the dynamic nature of human eye movements during visual exploration. Fig.1 illustrates and describes the human visual system that inspires us to design our model.

### 3.1 Model design and training

Akin to the human eyes [60, 61], our model uses retinal transformation [62] to generate separate inputs to WhereCNN and WhatCNN. For both, the retinal input consists of $64 \times 64$ samples non-uniformly distributed around the fixation. When describing a point in the retinal image and its corresponding point in the visual world in terms of the radial distance and the polar angle with respect to the fixation, their polar angles are the same while their radial distances are related by Eq. 1.

$$r = g(r') = \frac{b}{\sqrt{\pi}} \frac{1 - \exp(\ln(a)r'/2)}{1 - \exp(\ln(a)/2)} \tag{1}$$

where $r'$ and $r$ are the radial distances in the retinal and original images, respectively, $b$ is a constant that ensures $r_{\max}/g(r'_{\max}) = 1$, and $a$ controls the degree of center-concentration. Given a larger $a$, more retinal samples are closer to the fovea relatively to the periphery. We set $a = 15$ for WhatCNN and $a = 2.5$ for WhereCNN. In this setting, WhereCNN is more selective to global features, while WhatCNN is more selective to local features, mirroring the sampling bias of magnocellular and parvocellular retinal ganglion cells, as illustrated in Fig.1.

Both WhereCNN and WhatCNN use similar backbone architectures. The backbone consists of four blocks of convolutional layers. Each block includes two `Conv2D` layers (kernel size $3 \times 3$) followed by `ReLU` and `BatchNorm`. Applying $2 \times 2$ `MaxPool` between adjacent blocks progressively reduces the spatial dimension. The feature dimension following each block is 64, 128, 256, or 512. Atop this backbone CNN, both WhereCNN and WhatCNN use additional components to support different goals. For WhereCNN, the feature maps from the 3rd and 4th convolutional blocks are resized to

$16 \times 16$ and concatenated, providing the input to an additional convolutional block. Its output feature map is subject to `SoftMax` to generate a probability map of visual saliency. By random sampling by the probability of saliency, WhereCNN decides a single location for the next fixation. To avoid future fixations to revisit previously attended areas, inhibition of return (IOR) [63] is used. IOR keeps records of locations of prior visits as defined in Eq.2.

$$\textbf{IOR}(t) = \textbf{ReLU}\Big(1 - \sum_{\tau=1}^{t} G(\boldsymbol{\mu} = \boldsymbol{l}_{\tau}, \boldsymbol{\Sigma} = \sigma^2 \boldsymbol{I})\Big) \tag{2}$$

where $G(\boldsymbol{\mu}, \boldsymbol{\Sigma})$ is a 2D Gaussian function centered at $\boldsymbol{l}_{\tau}$ (prior fixations) with a standard deviation $\sigma$ at the $\tau$-th step. Its values are normalized so that its maximum equals 1. By applying the IOR to the predicted saliency map using an element-wise multiplication, the future fixations would not revisit the areas already explored.

For WhatCNN, the output feature map from the 4th convolutional block, after global average pooling, is given as the input to an additional layer of Gated Recurrent Units (GRU) [64], which recurrently update the representation from a sequence of fixations to construct a cumulative representation following another fully-connected layer.

We first pre-train each stream separately and then fine-tune them together through three stages.

**Stage 1 - WhereCNN**. We first train WhereCNN for image recognition using ILSVRC2012 [65] for object recognition and then fine-tune it for generating the saliency map to match human attention using SALICON dataset [66]. In this stage, we use random fixations to generate the retinal inputs to WhereCNN. Adam optimizer [67] (lr=0.002, $\beta_1$=0.9, $\beta_2$=0.99) is used with 25 epochs for SALICON training. At this stage, WhereCNN learns spatial attention from humans.

**Stage 2 - WhatCNN**. We first train WhatCNN for single-object recognition using ILSVRC2012 [65] and then fine-tune it for multi-object recognition using MSCOCO [68]. In this stage, we use the pre-trained WhereCNN to generate a sequence of eight fixations and accordingly apply the retinal transformation to generate a sequence of retinal inputs to WhatCNN for recurrent object recognition. Note that the training in Stage 2 is confined to WhatCNN, while leaving WhereCNN as pre-trained in Stage 1. Adam optimizer (lr=0.002, $\beta_1$=0.9, $\beta_2$=0.99) is used with 40 epochs for MSCOCO training.

**Stage 3 - WhereCNN & WhatCNN**. Lastly, we equally combine the two learning objectives to train both WhereCNN and WhatCNN altogether and end-to-end using eight fixations. Adam optimizer (lr=0.0002, $\beta_1$=0.9, $\beta_2$=0.99) is used with 25 epochs for training. In this stage, SALICON, which contain labels for both saliency prediction and object recognition, is used for training. More details can be found at Appendix B [1].

### 3.2 Model evaluation with human gaze behavior and fMRI responses

We use two criteria to evaluate how well a model matches the brain given naturalistic and dynamic visual stimuli. First, the model should generate similar human visual behaviors, such as visual perception and gaze behavior. Second, the model's internal responses to the stimuli should predict the brain's responses to the same stimuli through linear projection implemented as linear encoding models [69]. For our dual-stream model, we hypothesize that WhereCNN better predicts dorsal-stream voxels and WhatCNN better predicts ventral-stream voxels.

For this purpose, we use a publicly available fMRI dataset from a prior study [70], in which a total of 11 human subjects (4 females) were instructed to watch the movie Raiders of the Lost Ark (115 minutes) with unconstrained eye movements. The movie was displayed on an LCD projector with a visual angle of $17° \times 22.7°$. Whole-brain fMRI data was acquired in a 3-T MRI system with a gradient-recalled echo planar imaging sequence (TR/TE = 2.5s/35ms, flip angle = $90°$, nominal resolution = $3\text{mm} \times 3\text{mm} \times 3\text{mm}$). We preprocess the data by using the minimal preprocessing pipeline released by the Human Connectome Project (HCP) [71]

We test how well the model can predict the voxel-wise fMRI response to the movie stimuli through a learnable linear projection of artificial units in the model. To evaluate whether and how the two branches in the model differentially predict the two streams in the brain, we define two encoding

---

[1]The code is available at `https://github.com/minkyu-choi04/DualStreamBrains/`

models for each voxel: one based on WhereCNN and the other based on WhatCNN. We train and test the encoding models with data during different segments of the movie. To avoid overfitting, we apply dimension reduction to the internal responses in either WhereCNN or WhatCNN by applying principal component analysis (PCA) first to each layer and then to all layers while retaining 99% of the variance [30, 33]. We further convolve the resulting principal components with a canonical hemodynamic response function (HRF) that peaks at 5 seconds, down-sample them to match the sampling rate of fMRI, generating the linear regressors used in the encoding model. Using the training data (81% of the total data), we estimate the encoding parameters using L2-regularized least squares estimation. Using the held-out testing data (19%), we test the encoding models for their ability predicting the fMRI responses observed at each voxel and measure the accuracy of prediction as the correlation between the predicted and measured fMRI responses, denoted as $r_{where}$ and $r_{what}$ for the encoding models based on WhereCNN and WhatCNN. We test the significance of the prediction using a block permutation test [72] with a block size of 20-seconds and $100,000$ permutations and apply the false discovery rate (FDR) ($p < 0.05$). We further differentiate the relative roles of the brain's WhereCNN vs. WhatCNN in predicting the brain's dorsal and ventral streams for single voxels as well as regions of interest. For this, we define a relative performance (Eq.3).

$$p_{where} = \frac{r_{where}^2}{r_{where}^2 + r_{what}^2} \tag{3}$$

In the range from 0 to 1, $p_{where} > 0.5$ indicates better predictive performance by WhereCNN, while $p_{where} < 0.5$ indicates better predictive performance by WhatCNN.

### 3.3 Alternative models and control experiments

By design, the WhereCNN and WhatCNN branches within our model exhibit two key distinctions. WhereCNN is specifically trained to learn spatial attention by utilizing wider views, while WhatCNN focuses on object recognition through the use of local views. To explore the impact of input views and learning objectives on the model's capacity to predict brain responses, we introduce two modified control streams: ControlCNN-a and ControlCNN-b, designed as hybrid variants encapsulating diverse input views and learning objectives. ControlCNN-a receives a narrower view as its input and is trained for saliency prediction. Conversely, ControlCNN-b is curated to accommodate a broader view and primarily focuses its training on object recognition. These adjustments yield a versatile examination of their respective influences and functionalities. By combining WhereCNN or WhatCNN with ControlCNN-a or ControlCNN-b, we create four alternative dual-stream models (illustrated in Fig.4) and examine their abilities to explain the functional segregation of the brain's dorsal and ventral streams.

## 4 Results

### 4.1 WhereCNN learns attention and WhatCNN learns perception

The WhereCNN and WhatCNN branches in our model are specifically designed to fulfill different objectives: predicting human visual saliency and recognizing visual objects, respectively. In Fig.2, we present examples comparing human attention with the model's attention based on the SALICON's validation set. WhereCNN can successfully identify salient locations where humans are more likely to direct gaze. Additionally, WhereCNN can mimic human saccadic eye movements by generating a sequence of fixations that navigate the model's attention to those salient locations. In contrast, WhatCNN can recognize either single or multiple objects (macro F1 score on MSCOCO's validation set: 61.0).

### 4.2 WhereCNN and WhatCNN matches dorsal and ventral visual streams

By using linear encoding models, we use the WhereCNN and WhatCNN branches to predict fMRI responses during the processing of identical movie stimuli by both the model and the brain. Together, these two branches can predict responses across a wide range of cortical locations involved in visual processing. However, they exhibit distinct predictive power in relation to the dorsal and ventral streams. Generally, the WhereCNN branch exhibits superior predictive performance for the

WhereCNN learns human attention to mimic human gazes

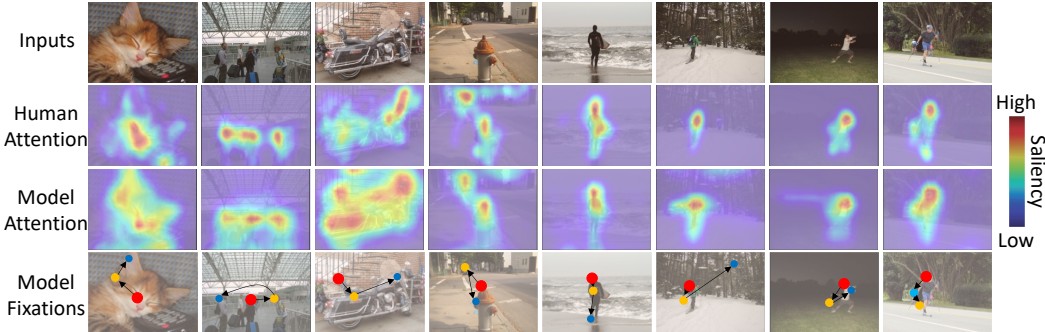

Figure 2: **Saliency prediction**. Given an image (1st row), WhereCNN generates a saliency map (3rd row) similar to the map of human attention (2nd row). Sampling this saliency map generates a sequence of fixations (as red/orange/blue circles in the order of time in the 4th row) similar to human saccadic eye movements (not shown).

(a) Relative contributions of WhereCNN and WhatCNN to prediction of brain responses

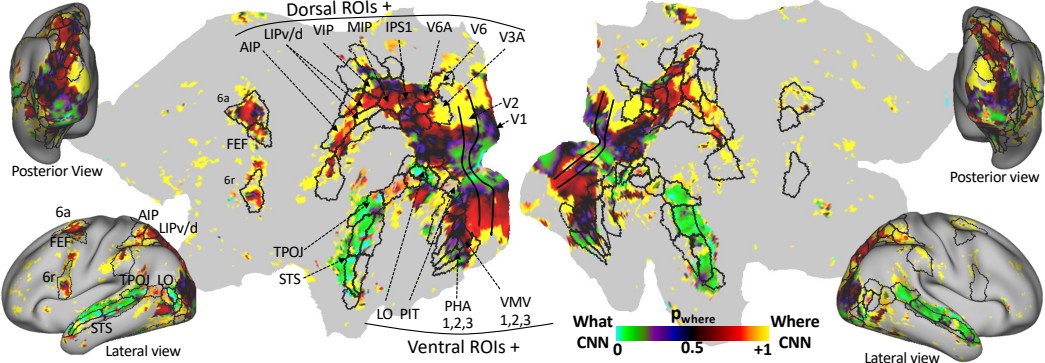

(b) Brain response predictability by WhereCNN vs WhatCNN

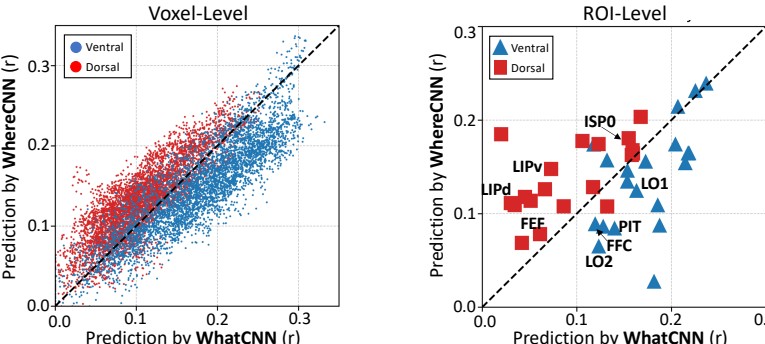

Figure 3: **Differential encoding of the dorsal and ventral streams**. (a) Relative contributions of WhereCNN and WhatCNN to the prediction of the fMRI response observed at each cortical location. Color highlights the locations significantly predictable by the model (FDR<0.05, block permutation test). The color itself indicates the degree by which WhereCNN is more predictive than WhatCNN (warm-tone) or the opposite (cool-tone). Visual areas are delineated and labeled based on brain altas [73]. Panel (b) plots the predictive performance by WhereCNN (y-axis) against that by WhatCNN (x-axis) and shows a clear separation of voxels (left panel) or ROIs (right panel) along the dorsal stream (red) vs. ventral stream (blue) relative to the dashed line of equal predictability. See Appendix A for the full ROI labels.

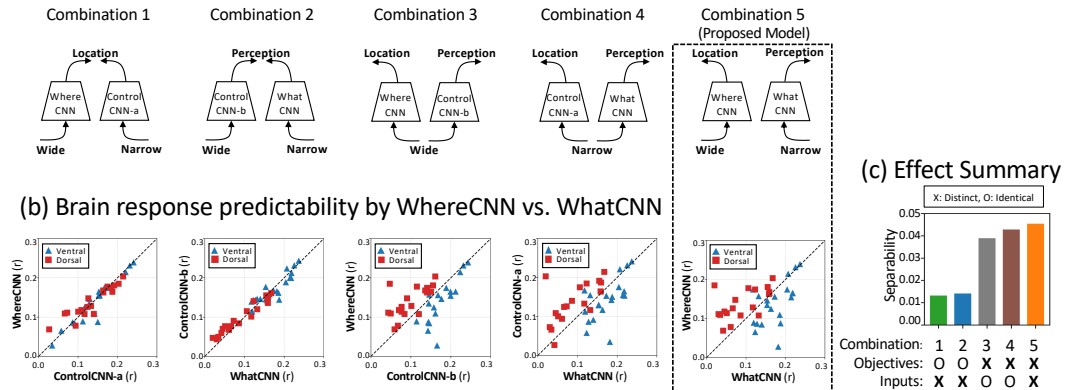

Figure 4: **Contributing factors of the dorsal-ventral functional segmentation.** (a) Alternative designs of the two-stream model for investigating the contributing factors of the functional segregation of two streams in the proposed model. In addition to WhereCNN and WhatCNN in the proposed model, we included ControlCNN-a (narrow input field of view for predicting location) and ControlCNN-b (wide input field of view for predicting perception) for an ablation study. (b) The predictive performances and functional segregations by the two streams are plotted for the dorsal (red squares) vs. ventral (blue triangles) ROIs for each of the alternative models in (a), correspondingly. The dashed diagonal lines represent equal predictive abilities of ventral and dorsal ROIs. (c) Quantitative evaluation of the functional segregation of the dorsal and ventral ROIs relative to the dashed line of equal predictability. Separability measures how far the predictions are away from the dashed line. Assuming the coordinates of a certain ROI is $(x, y)$, separability is calculated as the average of $|x - y|$ for all ROIs.

dorsal stream, while the WhatCNN branch performs better in predicting responses within the ventral stream (Fig.3). For the early visual areas (V1, V2, V3), WhereCNN better predicts the peripheral representations, while WhatCNN better predicts the foveal representations.

### 4.3 Factors underlying the functional segregation of the dorsal and ventral streams

We further investigate the underlying factors contributing to the model's ability to explain the functional segregation of the brain's dorsal and ventral visual streams (as depicted in Fig.3). Specifically, we examine the input sampling pattern and output learning objective, both of which are distinct for the WhereCNN and WhatCNN branches in our dual-stream model.

To investigate the contributing factors of the functional segregation in predicting human ventral and dorsal streams, we introduce four variations of the proposed model, where the two branches either share their inputs or have the same learning objectives, and compare their abilities to account for the functional segregation of the dorsal and ventral visual streams (as shown in Fig. 4). When the two branches solely differ in their input sampling, they are unable to explain the dorsal-ventral segregation (combination 1 and 2). However, when the two branches exclusively differ in their learning objectives, the functional segregation is better explained (combination 3 and 4). Moreover, when the two branches differ in both input sampling and learning objectives (combination 5), as utilized in our proposed model, the functional segregation is even more pronounced. These ablation experiments suggest that the distinct learning objectives of the brain's dorsal and ventral streams is the primary factor underlying their functional segregation.

### 4.4 Dual-stream: a better brain model than single-stream

We also compare our dual-stream model with single-stream alternatives. One of these alternatives is a baseline CNN that shares the same backbone architecture as a single branch in our dual-stream model. However, this baseline CNN is trained with original (224x224) images to recognize objects in ImageNet [65] and MS-COCO [68]. Thus, it serves as a direct comparison with either the WhereCNN or WhatCNN branch in our model. In addition, we also include AlexNet [14], ResNet18,

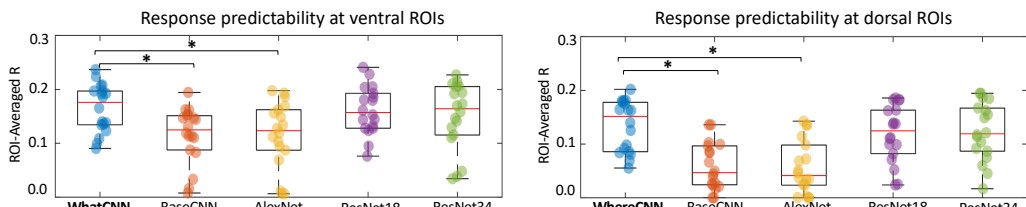

Figure 5: WhatCNN (left) or WhereCNN (right) vs. alternative single-stream CNNs. The boxplot shows the encoding performance of different models for ventral or dorsal visual areas. Each dot within the box plot signifies the average prediction accuracy r within a respective ROI in the ventral or dorsal region. Asterisk (*) represents a significant difference by the Wilcoxon signed-rank test ($\alpha = 0.05$).

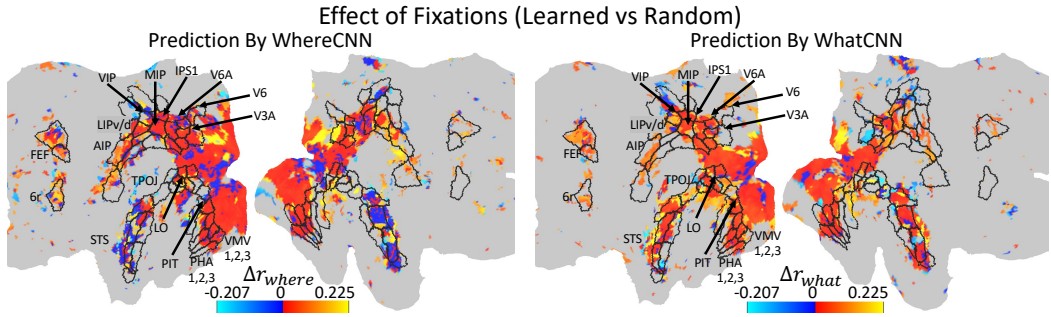

Figure 6: **Effects of attention-driven eye movements.** The use of attention to determine fixations vs. the use of random fixations is evaluated in terms of the resulting difference in the encoding performance by WhereCNN (left) and WhatCNN (right), denoted and color-coded as $\Delta r_{where}$ and $\Delta r_{what}$, respectively. Voxels displayed in warm colors indicate that the predictions are more accurate with learned fixations, whereas those in cool colors signify better predictions with random fixations.

and ResNet34 [15] as additional alternatives, which have been previously evaluated in relation to brain responses [30, 74]. We compare these single-stream alternatives with either branch in our model in terms of their ability to predict brain responses within dorsal or ventral visual areas (Fig.5). Despite their use of the same backbone architecture, the baseline under-performs WhatCNN in predicting responses in ventral visual areas, and under-performs WhereCNN in predicting responses in dorsal visual areas. This result suggests that the interactive and parallel nature of the dual-stream model renders each stream more akin to the functioning of the human brain, surpassing the performance of isolating a single stream. Moreover, WhatCNN or WhereCNN also performs better than AlexNet and comparably with ResNet18 and ResNet34, which are deeper than the architecture of our model.

### 4.5 Attention-driven eye movements improve encoding

Similar to human gaze behavior towards salient objects, our model learns spatial attention to guide fixations for parallel visual processing. In this study, we investigate whether and how the model's ability to predict brain responses depends on its utilization of attention-driven fixations. To examine this, we conduct experiments where the model is allowed to use either attention-driven fixations or random fixations to collect retinal samples, and we evaluate how this choice impacts the model's capability to predict brain responses by calculating $\Delta r = r^{learned} - r^{random}$ for all voxels, where $r^{learned}$ and $r^{random}$ are encoding performances of each voxel using learned or random fixations respectively. As depicted in Fig.6, employing attention-driven fixations leads to higher encoding accuracy by both WhereCNN and WhatCNN compared to the use of random fixations for a majority of visual cortical locations within both the dorsal and ventral streams.

# 5 Discussion

In summary, we introduce a new dual-stream neural network that incorporates the brain's mechanisms for parallel visual processing. The defining features of our model include 1) using retinal transformation to separate complementary inputs to each stream, 2) using different learning objectives to train each stream to learn either spatial attention or object recognition, and 3) controlling sequential fixations for active and interactive visual sensing and processing. We demonstrate that the combination of these features renders the model more akin to the human brain and better predictive of brain responses in humans freely engaged in naturalistic visual environments. Importantly, the two streams in our model differentially explain the two streams in the brain, contributing to the computational understanding as to how and why the brain exhibits and organizes distinct responses and processes along the structurally segregated dorsal and ventral visual pathways. Our findings suggest that the primary factor contributing to the dorsal-ventral functional segregation is the different goals of the dorsal and ventral pathways. That is, the dorsal pathway learns spatial attention to control eye movements [75, 7, 76, 77], while the ventral stream learns object recognition.

Although our model demonstrates initial steps to model parallel visual processing in the brain, it has limitations that remain to be addressed in future studies. For one limitation, the model uses different spatial sampling to generate the retinal inputs to the two streams but does not consider different temporal sampling that makes the dorsal stream more sensitive to motion than the ventral stream [38, 40, 39, 41]. For another limitation, the interaction between the two streams is limited to the common fixation that determines the complementary retinal input to each stream. Although attention-driven eye movement is an important aspect of human visual behavior shaping brain responses for both dorsal and ventral streams, the two streams also interact and exchange information at higher levels. The precise mechanisms for dorsal-ventral interactions remain unclear but may be important to understanding human vision or improving brain-inspired computer vision.

# 6 Acknowledgements

This research is supported by the Collaborative Research in Computational Neuroscience (CRCNS) program from National Science Foundation (Award#: IIS 2112773) and the University of Michigan.

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

# Appendix: A Dual-Stream Neural Network Explains the Functional Segregation of Dorsal and Ventral Visual Pathways in Human Brains

**Minkyu Choi[1], Kuan Han[1],**
**Xiaokai Wang[2], Yizhen Zhang[1,3], and Zhongming Liu[1,2]**

[1] Department of Electrical Engineering and Computer Science, University of Michigan, Ann Arbor, MI 48109
[2] Department of Biomedical Engineering, University of Michigan, Ann Arbor, MI 48109
[3] Department of Neurological Surgery , University of California, San Francisco, San Francisco, CA 94143
{cminkyu, kuanhan, xiaokaiw, zhyz, zmliu}@umich.edu

## A    Regions of Interests

In our study, we delineated our regions of interest (ROIs) into two primary segments: 1) the ventral visual stream and object recognition-related regions and 2) the dorsal visual stream and overt attention-related regions. This approach followed the parcellations proposed by [1]. For the dorsal visual stream, the ROIs includes V3A, V3B, V6, V6A, and V7. Within the parietal cortex, visuo-spatial information and overt attention are processed by the intraparietal sulcus (IPS) and the superior parietal lobule (SPL) [2, 3, 4, 5, 6]. The IPS encompasses V7, IPS1, IP0, IP1, and IP2; whereas the SPL consists of lateral intraparietal cortex (LIPv, LIPd), ventral intraparietal complex (VIP), anterior intraparietal (AIP), medial intraparietal area (MIP), 7PC, 7AL, 7Am, 7PL, and 7Pm. We also included the frontal eye field (FEF), which is acknowledged for controlling eye movements [7, 8, 9, 10]. In contrast, the ROIs associated with object recognition and the ventral visual stream encompassed V8, the posterior inferotemporal (PIT) complex, the fusiform face complex (FFC), and ventromedial visual (VMV) areas 1, 2, 3, along with the lateral occipital area (LO). In addition, we included the superior temporal sulcus (STS), which is recognized for processing multimodal signals, including auditory and visual cues [11, 12, 13]. Fig. S1 displays the full set of region labels, corresponding to Fig.3(a) from the main text. Among the parcellations by [1], regions including significantly predicted voxels either by the WhereCNN or WhatCNN are presented in Fig. S1.

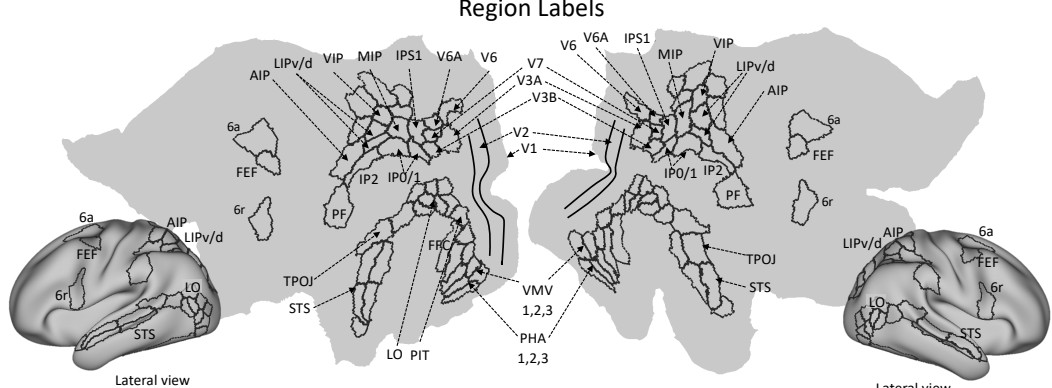

Figure S1: Region labels. Regions including significant voxels from Fig.3(a) in the main text are presented.

37th Conference on Neural Information Processing Systems (NeurIPS 2023).

# B    Training Details

The backbone Convolutional Neural Networks (CNNs) of both the WhereCNN and WhatCNN share the same architecture, consisting of four blocks of convolutional operations. Situated atop the backbone CNN, the WhereCNN and WhatCNN possess additional layers tailored to their specific objectives: the WhereCNN features two convolutional layers that produce 2D saliency maps, whereas the WhatCNN includes a Gated Recurrent Unit (GRU) layer followed by a fully connected layer for object classification.

During the pre-training of the backbone CNN, a global average pooling and a fully connected layer are integrated atop the backbone CNN, serving as a classifier. Upon completion of the pre-training process, the classifier is detached, allowing the pre-trained backbone CNN to be incorporated as a component of the WhereCNN or WhatCNN.

As detailed in Section 3.1 of the main text, our model underwent a three-stage training process. In this section, we will elaborate on the specifics of the pre-training phase.

**Stage 1 - WhereCNN** The backbone architecture of the WhereCNN was pre-trained on ILSVRC2012 [14] for an image classification task over 120 epochs. A batch size of $1,024$ was employed, along with the Adam optimizer [15] (lr=0.001, $\beta_1$=0.9, $\beta_2$=0.99). During pre-training, fixations for the retinal transformation were randomly generated across the image area. Once the backbone architecture had been pre-trained, we detached the classifier and initialized the WhereCNN using the model parameters obtained from the pre-training stage. We then performed SALICON training, as described in Section 3.1 of the main text.

**Stage 2 - WhatCNN** In a process mirroring Stage 1, the backbone of the WhatCNN was also pre-trained on ILSVRC2012 [14] for an image classification task over 120 epochs, utilizing random fixations and the Adam optimizer (lr=0.001, $\beta_1$=0.9, $\beta_2$=0.99). After pre-training the backbone CNN, we initialized the WhatCNN using the weights of the pre-trained backbone CNN.

Subsequently, the WhatCNN, initialized with the pre-trained weights as a whole, was trained on ILSVRC2012 [14] for object recognition using four fixations. Four randomly generated fixations were employed for training the WhatCNN for 55 epochs, again utilizing the Adam optimizer (lr=0.001, $\beta_1$=0.9, $\beta_2$=0.99). After this stage, we conducted a fine-tuning process using the learned fixations from the WhereCNN. In this stage, the WhereCNN, after the pre-training in Stage 1, was incorporated to guide the WhatCNN's fixations. However, only the WhatCNN was optimized, while the WhereCNN remained unchanged. This fine-tuning with learned fixations deployed four gazes, utilizing the Adam optimizer (lr=0.0001, $\beta_1$=0.9, $\beta_2$=0.99) over 25 epochs. Finally, the WhatCNN underwent further training on MSCOCO, as described in Section 3.1 of the main text.

**Stage3 - WhereCNN & WhatCNN** During this stage, both WhereCNN and WhatCNN, trained in the previous stages, were used to initialize model weights, followed by further end-to-end training, leveraging the stream-specific objectives (object recognition and saliency prediction, respectively). As the training requires labels for both tasks, the model was trained using images in the SALICON dataset, which contain labels for both saliency prediction and object recognition.

The model samples fixations from the predicted saliency maps from WhereCNN. As this sampling process is non-differentiable, the gradients from object recognition cannot optimize the weights of WhereCNN. To tackle this issue, we utilized REINFORCE [16] to approximate the gradient for WhereCNN. At the time $t$, a fixation $l_t$ is generated by WhereCNN, based on which WhatCNN predicts a class prediction $p_t$. Then, in the context of REINFORCE, the reward $r_t$ of choosing $l_t$ as the fixation is calculated as the reduced classification loss relative to the previous time step $r_t = CE(p_{t-1}, \text{label}_c) - CE(p_t, \text{label}_c)$, where $CE$ is the cross-entropy loss, $\text{label}_c$ is class labels. The goal of REINFORCE is to maximize the discounted sum of rewards, $R = \sum_{t=1}^{T} \gamma^{t-1} r_t$, where $\gamma \in (0, 1)$ is the discount factor and set as 0.8.

In this stage, we strived to minimize the object recognition and saliency prediction losses while maximizing the discounted sum of rewards. As indicated in Section 3.1 of the main text, we utilized the Adam optimizer (lr=0.0002, $\beta_1$=0.9, $\beta_2$=0.99) for 25 epochs for this training stage.

**For All Stages** All training stages were conducted using four NVIDIA A40 GPUs. All codes are written in Pytorch 1.9.1.

## C   Saliency Maps and Inhibition of Returns

Once the saliency maps were generated by WhereCNN, inhibition of return (IOR) was used to prohibit future fixations to re-visit image areas that had been already explored. This process is illustrated in Fig. S2

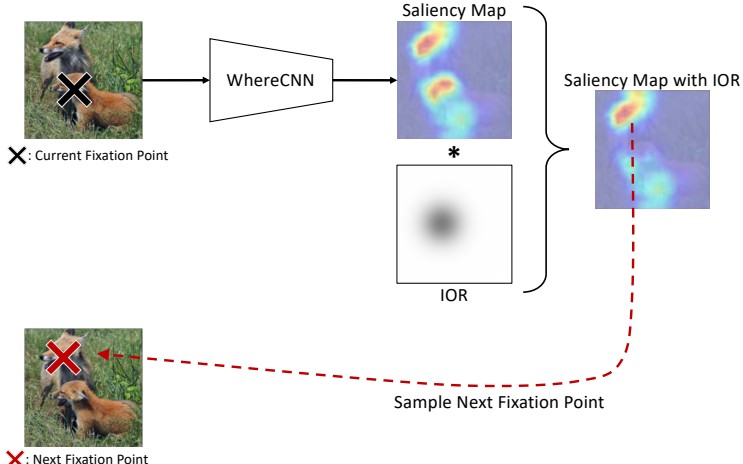

Figure S2: Process of determining the next fixation point given the current fixation. A saliency map, generated by WhereCNN, is multiplied element-wise (indicated by ∗) with the inhibition of return (IOR) to prevent future fixations from reverting to previous positions. In the IOR, white and black colors correspond to values of $1$ and $0$, respectively.

In the process of determining the next fixation, the WhereCNN generate a saliency map based on the current fixation. The location of this subsequent fixation is guided by the saliency map's probabilistic distribution. However, it's important to note that if the current fixation point possesses a high probability, subsequent fixations are likely to occur in proximity to the present fixation.

To ensure a more dynamic and comprehensive exploration of the visual field, we employed the principle of Inhibition of Return (IOR), detailed in Eq.2 of the main text, and presented again here in Eq.4.

$$\mathbf{IOR}(t) = \mathbf{ReLU}\Big(1 - \sum_{\tau=1}^{t} G(\boldsymbol{\mu} = \boldsymbol{l}_\tau, \boldsymbol{\Sigma} = \sigma^2 \boldsymbol{I})\Big) \tag{4}$$

where $G(\boldsymbol{\mu}, \boldsymbol{\Sigma})$ is a 2D Gaussian function centered at $\boldsymbol{l}_\tau$ (prior fixations) with a standard deviation $\sigma$ at the $\tau$-th step. The Inhibition of Return (IOR) is initially created at a resolution of $224 \times 224$ with $\sigma = 25$, and subsequently resized to align with the dimensions of the saliency map. IOR serves to decrease the saliency of previously attended areas, thereby preventing the model from repetitively focusing on these regions. This mechanism is informed by the model's all prior fixation history. The IOR map is designed such that it assigns lower values (approaching $0.0$) in the vicinity of prior fixation points, and higher values (up to $1.0$) in regions further away. Thus, when the IOR map is element-wise multiplied with the saliency map, it effectively reduces the saliency values in areas already explored.

Following the application of IOR, the subsequent fixation point is decided upon by considering the adjusted saliency map. It is then chosen based on the probabilistic distribution within this updated map. This strategy encourages more diverse fixations and facilitates a broader and more comprehensive understanding of the scene.

## D   WhereCNN's Saliency Maps and Fixation Points

The original images are presented in Cartesian coordinates. Once the retinal transformation is applied to these images, the resultant retinal images adopt retinal coordinates, as detailed in Eq.1 of the main

text. Since the inputs to the WhereCNN operate in retinal coordinates, it naturally follows that the output saliency maps mirror this coordinate system. To visualize these within this paper, we utilize the inverse function of Eq.1, thereby transforming the saliency maps from retinal back to Cartesian coordinates.

In preparation for our model's processing of the movie *Raiders of the Lost Ark*, we reduce the frame rate to 6 frames per second (fps). This adjustment helps mitigate computational and memory costs associated with the handling of the extracted features. As the model engages with the movie, a solitary fixation point is established for each frame. Importantly, the Inhibition of Return (IOR) mechanism is not invoked during the model's interaction with the movie. Fig. S3 showcases saliency maps and fixation points derived from segments of the movie *Raiders of the Lost Ark*. Frames situated on the same horizontal axis are selected at a rate of 1 fps.

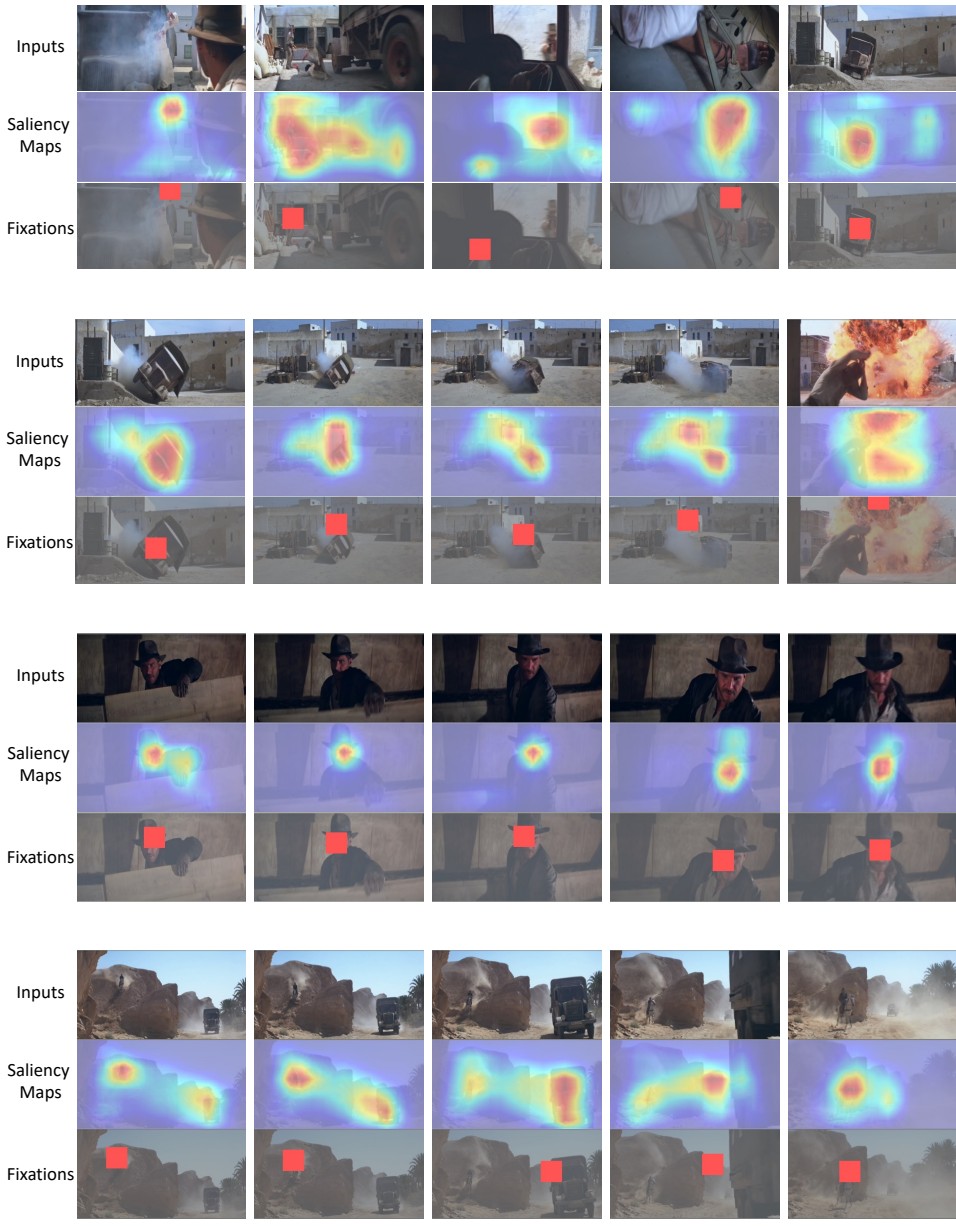

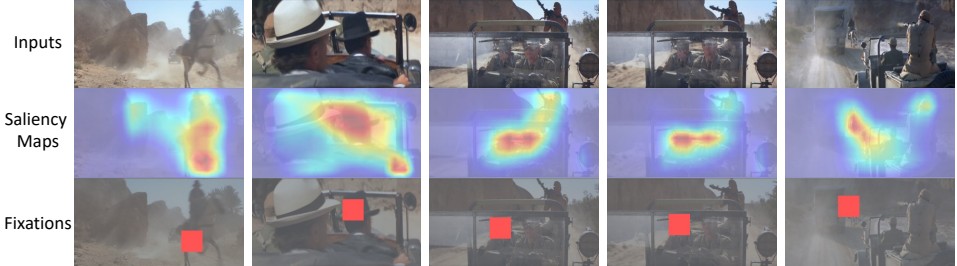

Figure S3: Given the movie frames (1st row), the WhereCNN generates saliency maps (2nd row) and fixations (3rd row). The red marker in the 3rd row presents the fixation point.

# E  Investigating Layer-wise Correspondence to Visual Cortex

In the main text, the whole features from the all layers of each stream are used for predicting voxel activities (noted as Stream-wise encoding). In an alternative way, the features from each layer can be used to predict voxel activities, instead of concatenating all the layers, (noted as Layer-wise encoding). In this way, the hierarchical correspondence between each layer in the model to the ROIs of the visual system can be observed.

With the layer-wise encoding scheme, we predicted fMRI responses using features from each layer in the WhereCNN and WhatCNN. Fig. S4 associates each voxel to one (color-coded) layer most predictive of that voxel for either (a) WhatCNN or (b) WhereCNN. Fig. S4 (a) shows that the lower layers of WhatCNN better predict earlier visual areas such as V1/V2, whereas the higher layers of WhatCNN better predict higher-order visual areas such as LO and PIT, consistent with prior studies [17, 18]. The results with the WhereCNN show different patterns, as shown in Fig. S4 (b). Within early visual areas, the lower layers of WhereCNN better predict foveal representations, whereas the higher layers better predict peripheral representations.

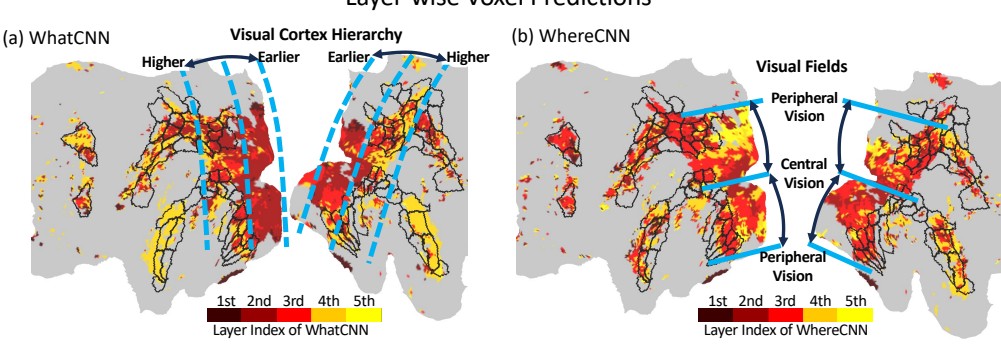

Figure S4: Each voxel is predicted by the features from a single layer from (a) WhatCNN and (b) WhereCNN. Layer indexes are color-coded so that the layer best predicting each voxel is presented.

# F  Implications to the Computer Visions

In the current study, we demonstrated that the biologically plausible components (two stream, retinal sampling and eye movements) can be used to build a better model for the human visual cortex in a naturalistic viewing condition. At the same time, those components we considered in this study may also bring benefits to the computer vision applications.

1) **Efficiency**. Unlike conventional CNNs that process entire images, our dual-stream model allows serial processing. It concentrates processing power on key image regions through attention directed fixations. This serial processing may significantly lower memory and computational overhead,

because resources are allocated only to the crucial image regions. It is plausible that such efficiency underpins the brain's adoption of dual stream processing due to biological constraints on energy use.

2) **Adaptability**. The dual streams of our model offer complementary lenses for visual exploration and perception in real-world environments. One stream provides a broad yet rough overview of the environment. The other gathers detailed observations with precision. Their synergistic interaction may facilitate adaptive behaviors for tasks like visual search, object detection in complex and cluttered scenes. Moreover, the distinct functions of each of the parallel streams present a combinatorial flexibility when leveraged together, potentially enhancing the model's overall capability to adapt to diverse visual challenges, including potential applications in robotics.

However, leveraging such potential benefits within the scope of current study face challenges. First, mainstream datasets like ImageNet and MS-COCO offer a narrow view and lack the high-resolution detail our model thrives on. Moreover, these datasets often focus on large, central objects, limiting our model's adaptability that benefits object recognition. A better benchmark to our model would be high-resolution panoramic images or synthetic virtual reality environments to accommodate unlimited fixation variances. In such settings, the efficiency and adaptability of our model should be more appealing.

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
