# OpenReview forum: "A Dual-Stream Neural Network Explains the Functional Segregation of Dorsal and Ventral Visual Pathways in Human Brains"
_NeurIPS.cc/2023/Conference — NeurIPS 2023 poster_

### Official Review · Reviewer_xqDS · 2023-07-05

**Soundness:** 3 good
**Presentation:** 3 good
**Contribution:** 3 good
**Rating:** 7
**Confidence:** 4

**Summary:**

This paper analyzes which ANN structures are more suitable for conversion to SNN. Based on the analysis results, the RepConv model is chosen as the baseline ANN model. The ANN-to-SNN conversion is performed using Value-range (VR) encoding and Clamp and Quantization (CQ) training, resulting in a high-performance and low-latency SNN network.

**Strengths:**

1.	The neural network model structure and training approach are novel. Compared to previous dual-stream models, this work trains the two streams separately using different tasks and combines their training by incorporating changes in attention, enabling simultaneous training.
2.	The network successfully models the functional specialization of the ventral and dorsal visual pathways and demonstrates that the two streams correspond to different visual pathways. Additionally, it exhibits comparable predictive performance to other single-pathway neural networks.
3.	The ablation experiments are comprehensive and provide insights into the importance of learning objectives and retinal input in the functional specialization of the human visual pathways.
4.	The content of the paper is substantial, the experimental results are solid, and the writing is good.


**Weaknesses:**

1.	Some details need to be addressed. What datasets were used for training the two streams synchronously? Why is the learning rate lower when training the two streams together compared to training them separately?
2.	Although the network has a recurrent structure and multiple time steps, the training dataset consists of static data. However, fMRI visual input is dynamic. Could dynamic datasets be considered for training the network?
3.	Single-pathway networks with a single training objective (object recognition) such as ResNet18 and ResNet34 also exhibit good predictions of fMRI data for both visual pathways. However, intuitively, such models should not be able to predict the dorsal visual pathway well. Does this result indicate that there is no significant difference in fMRI data features between the ventral and dorsal pathways?
4.	When using network features to predict fMRI data, is it possible to predict separately for each layer to examine whether each layer corresponds to different regions of the two pathways?


**Questions:**

See the weaknesses

**Limitations:**

Some experimental details and explanations are needed.

---

> ### Author Rebuttal · Authors · 2023-08-10
>
> We are grateful for your positive evaluation, constructive feedback, and insightful comments.
> ***
> 1\. Regarding the questions on why the learning rate for stage3 was set lower than that for stages 1 and 2, the answer is that our empirical findings led us to this decision, as a reduced learning rate during stage3 resulted in a more stable learning curve. This is also a common strategy used in multi-task / multi-stage / transfer learning schemes in deep learning.
>
> For training stage 3, we used the SALICON dataset for training. Stage 3 utilizes both objective functions of saliency prediction and object recognition. Because the images in SALICON are collected from the MS-COCO, images in SALICON have labels for saliency prediction and object recognition. This is the major reason that we use SALICON as the training dataset in stage 3. We have also included more training details in the global response section. In light of the reviewer’s suggestion, we will include substantial training details in Appendix B for the final version of this paper.
>
> ***
>
> 2\. We thank the reviewer for the suggestion. We agree that training our model on dynamic datasets instead of static datasets, particularly for tasks like action recognition, would be a feasible and compelling idea, and may result in a better alignment with the fMRI data collected with dynamic visual input. Yet, a significant challenge is the scarcity of dynamic datasets for saliency prediction. The prevalent use of static images in most saliency prediction datasets have unfortunately limited our choice to use static images for model training.
>
> ***
> 3\. Concerning the comment on Figure 5, we would like to firstly acknowledge an error in the figure's label. The current Figure 5 in the main text presents the two panels for WhereCNN / dorsal ROIs (left panel) and WhatCNN / ventral ROIs (right panel). However, The left and the right labels were switched from each other, meaning that the left panel is about WhatCNN / ventral ROIs, and the right panel is about WhereCNN / dorsal ROIs. The revised figure has been appended to the PDF in the global response, and this change does not affect the other analysis or conclusions we made. We deeply regret this oversight.
>
> From a neuroscience perspective, it is well-established that the dorsal and ventral streams are selective to different features of visual stimuli, each having unique roles. For instance, while the dorsal stream deals with spatial layouts and it is sensitive to motion, the ventral stream extracts object features [1, 2]. Thus, we expect that the two streams show divergent representations in fMRI responses to movies.
>
> The updated Figure 5 illustrates that baseline models such as BaseCNN, AlexNet, ResNet18, and ResNet34 better predict the ventral ROIs than the dorsal ones. Several dorsal ROIs, with r values nearing 0, suggesting a lack of predictability. In contrast, our WhereCNN and WhatCNN models provide superior predictions across most dorsal and ventral ROIs, outperforming both the baseline models of comparable size like BaseCNN and AlexNet and the larger ones such as ResNet18 and ResNet34.
>
> ***
> 4\. We appreciate the reviewer suggestion on layer-wise encoding. In light of this suggestion, we predicted fMRI responses using features from each layer in the WhereCNN and WhatCNN. The results have been included in the global response as Figure R4. Briefly, Figure R4 associates each voxel to one (color-coded) layer most predictive of that voxel for either (a) WhatCNN or (b) WhereCNN. Figure R4 (a) shows that the lower layers of WhatCNN better predict earlier visual areas such as V1/V2, whereas the higher layers of WhatCNN better predict higher-order visual areas such as LO and PIT, consistent with prior studies [3, 4]. The results with the WhereCNN show different patterns, as shown in Figure R4 (b). Within early visual areas, the lower layers of WhereCNN better predict foveal representations, whereas the higher layers better predict peripheral representations. However, the layer-wise encoding of higher visual areas seems not readily explainable. We need more time for deeper analysis before making an affirmative conclusion. In our provisional interpretation, the lack of a hierarchy of layer-wise prediction in WhereCNN, compared to WhatCNN, might suggest that the representational hierarchy in the visual cortex is likely more attributable to visual perception/recognition rather than attention.
>
> ***
> __References__
>
> [1] Haxby, J.V., Grady, C.L., Horwitz, B., Ungerleider, L.G., Mishkin, M., Carson, R.E., Herscovitch, P., Schapiro, M.B. and Rapoport, S.I., 1991. Dissociation of object and spatial visual processing pathways in human extrastriate cortex. Proceedings of the National Academy of Sciences, 88(5), pp.1621-1625.
>
> [2]  Mishkin, M., Ungerleider, L.G. and Macko, K.A., 1983. Object vision and spatial vision: two cortical pathways. Trends in neurosciences, 6, pp.414-417.
>
> [3] Wen, H., Shi, J., Zhang, Y., Lu, K.H., Cao, J. and Liu, Z., 2018. Neural encoding and decoding with deep learning for dynamic natural vision. Cerebral cortex, 28(12), pp.4136-4160.
>
> [4] Cadena, S.A., Denfield, G.H., Walker, E.Y., Gatys, L.A., Tolias, A.S., Bethge, M. and Ecker, A.S., 2019. Deep convolutional models improve predictions of macaque V1 responses to natural images. PLoS computational biology, 15(4), p.e1006897.

---

> > ### Comment · Reviewer_xqDS · 2023-08-16
> >
> > I appreciate the authors' detailed reply. I would like to keep the score.

---

> > > ### Author Response · Authors · 2023-08-17
> > > **Rebuttal by Authors**
> > >
> > > Thank you for recognizing our detailed reply. We welcome any further comments.

---

### Official Review · Reviewer_E5bU · 2023-07-05

**Soundness:** 3 good
**Presentation:** 4 excellent
**Contribution:** 3 good
**Rating:** 7
**Confidence:** 4

**Summary:**

The main contributions is a dual stream neural network to model human visual processing. The network is composed of two CNNs (where and what) to model the dorsal and ventral pathways. The authors demonstrate that they can predict human gaze behavior and neural responses using their model, and gain new insights in to the role of the dorsal stream and how it interacts with the ventral stream.

**Strengths:**

A dual stream CNN to model the dorsal and ventral stream is novel as far as I know and quite interesting. The authors do a good job of the motivating their work and being clear about their contributions. As they note, many approaches to comparing human and deep neural networks stop at classification CNNs, so moving beyond simple object recognition and separating out different streams of processing is an interesting contribution.

In addition, the experiments appear to be well reasoned. The authors create a model motivated by properties of the human brain, and provide logical comparisons points --the authors provide both an eye-gaze and fMRI analysis to compare their model to humans and also provide a baseline single stream CNN to make it clear what benefits arise from the dual stream design.  Overall, the work is very clear.

**Weaknesses:**

The work does not have enough discussion of how their dual stream model could be useful to a more broad computer vision audience. Although the authors show their dual stream model does better than single stream at explaining brain response, the work would be strengthened by more clearly demonstrating the use for a dual stream system directly on a computer vision tasks (i.e. does object recognition or detection improve? or robustness?).

The authors only consider CNNs and supervised training in their modeling. While these models are often used when comparing humans and deep neural networks, they are not top performers in the context of computer vision. Even in the context of biological modeling, more specialized models (for example, adversarially trained models) are thought to share more similarities with human visual representations than  standard CNNs.

**Questions:**

As I mentioned in weakness, could the authors touch on how their dual stream model is relevant a wider computer vision audience? What are potential performance benefits of the dual stream approach and/or some the gaze mechanisms you incorporated (better accuracy? robustness? efficiency?). Numerical results would be appreciated, but general statements/discussions are also fine.

What is the size of your dual stream model? How costly is it to train? Could you point me to performance scores? I would like to know more of these details about the model.

On prior work, could the authors touch on how their work relates to metrics like BrainScore (Schrimpf et al. 2018)? Also, how does your model compare to GistNet (Wu et al. 2018) which I believe takes a two pathway approach.

**Limitations:**

Yes, the authors adequately address limitations. The authors note that their model does not address temporal sampling and does not completely capture how the dorsal and ventral streams interact with one another. I might argue that relying on small CNNs for their model and relying on tasks like recognition and detection to learn visual representation could be limiting, but I discuss these concerns in weaknesses and questions and find the work good otherwise.

---

> ### Author Rebuttal · Authors · 2023-08-10
>
> We are glad that the reviewer recognizes the novelty, clarity, and contribution of our paper and offers constructive comments. Our responses to the weaknesses and questions are as below.
>
> ***
> __Applications in computer vision__
>
> The main focus of this paper is on modeling the human visual system, including its ventral and dorsal streams. We agree with the reviewer that demonstrating the benefits of our dual-stream model for a range of computer vision applications could enhance our work’s appeal. Given the page limit, we opt to keep our primary focus on computational neuroscience and will perform a comprehensive extension of the work for computer vision applications in a different paper. That said, we will include more discussions about several potential advantages for computer vision and the remaining challenges.
>
> __1) Efficiency.__ Unlike existing models that process entire images, our dual-stream model allows serial processing. It concentrates processing power on key image regions through attention directed fixations. This serial processing may significantly lower memory and computational overhead. It is plausible that such efficiency underpins the brain’s adoption of dual stream processing due to biological constraints on energy use.
>
> __2) Adaptability.__ The dual streams of our model offer complementary lenses for visual exploration and perception in real-world environments. One stream provides a broad yet rough overview of the environment. The other gathers detailed observations with precision. Their synergistic interaction may facilitate adaptive behaviors for tasks like visual search, object detection in complex and cluttered scenes. Moreover, the distinct functions of each of the parallel streams present a combinatorial flexibility when leveraged together, potentially enhancing the model’s overall capability to adapt to diverse visual challenges, including potential applications in robotics.
>
> However, leveraging such potential benefits within the scope of current study face challenges. First, mainstream datasets like ImageNet and MS-COCO offer a narrow view and lack the high-resolution detail our model thrives on. Moreover, these datasets often focus on large, central objects, limiting our model's adaptability that benefits object recognition. A better benchmark to our model would be high-resolution panoramic images or synthetic virtual reality environments to accommodate unlimited fixation variances. In such settings, the efficiency and adaptability of our model should be more appealing.
>
> ***
> __Concerns on CNNs and supervised learning__
>
> Thank you for the insightful comments. Our use of CNN and supervised learning in this study is motivated by a comprehensive literature on the similarity of CNNs and brain mechanisms in vision. However, we agree with the reviewer that our central idea is not limited to using CNNs exclusively as the architecture. For example, Transformer based architectures are increasingly are potential alternatives to CNNs. However, the biological relevance of Transformer, although hinted at in recent studies, remains unclear to us. Given our primary interest in brain-inspired vision models, our current position is still to prefer CNNs for this paper. Nevertheless, we concur with the reviewer’s suggestion that exploring a diverse set of architectures in dual-stream visual processing merits further investigations.
>
> Likewise, supervised learning is not the only strategy for training dual-stream vision models. The recent advancement in unsupervised and self-supervised learning is encouraging. Such approaches align with our aspirations in brain-inspired learning for vision. As of now, we have not yet pinpointed an effective self-supervised learning strategy tailored for dual-stream models. Moreover, we are amenable to exploring other learning strategies, such as adversarial training (mentioned by the reviewer). We also value any further recommendations from the reviewer. In the final version of this paper, we will add discussions about alternative architectures and learning strategies while citing related literature.
>
> ***
> __Size and performance__
>
> Our model (including both streams) comprises 4.73 million trainable parameters, and performs multi-label classification on the test images of MS-COCO by F1 score 61.0. We recognize the importance of demonstrating the advantages of our dual-stream model to the broader computer vision community. As elaborated in our response to another comment, the datasets (ImageNet or MS-COCO) used in this study are not ideal for demonstrating the strengths of our model for computer vision. We acknowledge this weakness.
>
> ***
> __BrainScore__
>
> Your suggestion about BrainScore is valuable. We agree with the premise. Indeed, brain-like vision models should be evaluated not just based on their performance on computer vision tasks but also their capability to predict neural responses and behavioral measures. While we have yet to submit our model to BrainScore, we are keen on leveraging BrainScore in the future. However, we anticipate certain challenges. For example, the neural data used by BrainScore are with central fixations, while our model actively adjusts its fixations. We prefer dynamic and naturalistic stimuli, while BrainScore mainly uses static images for its evaluations.
>
> ***
> __GistNet__
>
> Thanks for referring us to GistNet. We will discuss this work in the section of prior work. Indeed, our model and GistNet both utilize a dual-stream architecture. The primary distinction is our model’s active exploration of images by varying fixations, while GistNet assumes fixations centered on objects. Relevant to this distinction is our intended use of WhereCNN to learn spatial attention and use attention to drive eye movements.

---

> > ### Comment · Reviewer_E5bU · 2023-08-19
> > **response to author rebuttal**
> >
> > Thank you for your thoughtful response to my concerns. Most of my issues have been addressed, so I raised my score

---

> > > ### Author Response · Authors · 2023-08-19
> > > **Rebuttal by Authors**
> > >
> > > We are glad that our responses have addressed most of your concerns. Thank you for your feedback and recognition.

---

### Official Review · Reviewer_7ncd · 2023-07-06

**Soundness:** 3 good
**Presentation:** 3 good
**Contribution:** 2 fair
**Rating:** 5
**Confidence:** 4

**Summary:**

This paper focuses on investigating differences between the human ventral and dorsal visual streams. The authors develop a dual-stream model that has different retinal inputs to support “what” processing (object recognition) or “where” processing (guiding the fixations). The authors then test how well the model activations are able to predict human fMRI activity and see a distinction where the “What” model better predicts ventral visual areas and the “Where” model better predicts dorsal areas.

**Strengths:**

* Although there is a large amount of work studying the similarity of image-computable models and the ventral visual stream, there is less work done studying the dorsal stream. This paper provides an interesting analysis of the functional separation of these two pathways.
* The retinal framework and its relation to biology is original and significatn, as well as the model leading to different “fixations”.
* The dual-task model presented is original, and the experiments on different versions of the model provide interesting interpretation of the training components.


**Weaknesses:**

* In the abstract and elsewhere (ie line 90 and line 248-250), the authors state that “the functional segregation of the brain’s dorsal and ventral visual pathways is driven primarily by their different learning objectives”. This is too strong of a claim and not supported by the results (a correlation $\neq$ causation fallacy), as there is no evidence presented in the paper showing that the brain itself relies on these specific learning objectives.
* The “WhereCNN” is initially trained on Image Recognition and then fine-tuned on saliency maps. This makes the differences between the two models difficult to interpret, as it is *also* optimized for object recognition.
* The authors detail the interesting addition of foveated vision and retinal transformations, however from the results it appears that this is less necessary for performance than the two different tasks, but the relevance of this to our understanding of biology is not discussed. It is hard to say whether this lack of a result is due to the retinal sampling being incorrect, or if it simply doesn’t matter for this level of analysis.


**Questions:**

a)	In Figure 3b, it looks like there are many ventral voxels that are predicted better with the “Where” CNN. In contrast, there are few dorsal voxels that are predicted well by the “What” CNN. Is there an interpretation for this? And are voxels from early visual areas included in this plot (if so, what color are they?)

b)	The colorbar on Figure 3a is misleading because there is little difference between the darkest blue and the darkest red. A diverging color map would better capture the data.

c)	There are control experiments presented with alternative models, but from what I can tell these alternative models always have “dual streams”. Is it necessary to have the two streams to get good predictivity for these voxels or could it be captured with a single model performing two tasks (especially given that the inputs seem to matter less than the task)?

d)	A fourth model with different inputs but predicting “perception” may show a different trend, and should possibly be included for completeness. Similarly a model with only wide inputs to each of the architectures.

e)	Additional details of training, or a direct reference to the supplement where these details are provided, would be helpful. Specifically, on line 165 it is stated that the WhereCNN “learns spatial attention from humans”. Is this separate training than the SALICON dataset? And the training dataset, updates/labels etc are not given for Stage 3 of training.

f)	Quantification of the similarities between human and model attention/saccadic eye movements would be nice. Also, are the images in Figure 2 from the train or test dataset?

g)	Are the human eye movements during the fMRI activity recorded and available? If so, how well do these match up with those predicted by the model? (Maybe beyond the scope but an interesting analysis).

Minor:

i) Section 4.1 seems less of a result and more like validation that the training worked, as these validations are what the networks were trained to do.

ii) Lines 99-100 are technically true (in that the models do not fully replicate the way human eyes sample visual inputs) however neither does this study (it is still a model and does not capture all aspects of the eye). It would be better to explicitly state the differences between the previous studies and the model used here.

iii)	Clarifying Figure 4 and its caption to make it clear that the right most model is the standard model used in the rest of the paper would be helpful.

iv)	On line 265 there is a comment about WhatCNN and WhereCNN performing comparably to deeper models, but I don’t understand why depth matters in this context?



**Limitations:**

Limitations are addressed in the final paragraph of the discussion.

---

> ### Author Rebuttal · Authors · 2023-08-10
>
> We are glad that you recognize some strengths, interests, and novelties of our work. We also respect your critical evaluation and feedback.
>
> ***
> 1. __Overreach claims.__ We respect your concern regarding the potential overreach of our initial claim. We have revisited our wording to more accurately reflect our findings. Our revised statement now reads, “By evaluating the linear alignment between dual-stream models and brains during dynamic natural vision, we infer that the distinct responses and representations of the ventral and dorsal streams are more influenced by their distinct goals in visual attention and object recognition than by their specific bias or selectivity in retinal inputs.” Indeed, we do not present direct evidence concerning the brain’s learning processes. Our approach centers on reverse engineering the brain using neural network models. We embed hypotheses in these models and then test, compare, or falsify the hypotheses by comparing these models to the brain. While this method is correlational in nature, it is a valid inference framework established in the computational neuroscience community. In light of your concern, we will carefully reword our claims and findings in the final version of this paper.
>
>
> 2.  __Pre-training concerns.__ Indeed, both streams were initialized by pre-training with the same task of image classification due to practical considerations. Due to the small size of the human saliency dataset, training WhereCNN without pre-training would be challenging. Following pre-training, subsequent training phases train both streams on consistently different tasks (spatial attention for WhereCNN, object recognition for WhatCNN). Although both streams start with the same pre-training task, their resulting representations post-training diverge (Figure 3) suggesting that the two streams have learned to build distinct representations through their respective learning objectives.
>
>
> 3. __Retinal inputs vs. functional goals.__ The parallel dorsal vs. ventral visual streams have two major distinctions. They receive separate retinal inputs and serve different functional goals, for which we focus on spatial attention vs. visual perception in this paper. By accounting for these distinctions in our dual-stream model, we show that the functional segregation of the dorsal and ventral streams is more attributable to their divergent goals than their varied inputs. This finding is valid at least for the level of our analysis with fMRI data during movie viewing. However, we do not imply that foveated vision or retinal transformation is not important for vision in general. We recognize that prior studies have shown the utility of modeling the retina for adversarial robustness, neural activity control, to name a few [Vuyyuru et al., 2020, Bashivan et al., 2019].
>
> ***
> a) Figure 3b. Voxels and regions in Figure 3b exclude early visual areas, which are involved in both dorsal and ventral pathways. That said, WhatCNN and WhereCNN can capture the differentiation between foveal or peripheral representations in early visual areas (Figure 3a). While a significant portion of the ventral and dorsal streams align well with the predictions of the WhatCNN and WhereCNN, we concur that there are (many fewer) instances of misalignment or overlap. Pinpointing a single reason for this observation is challenging. A plausible interpretation is that there is shared information in the dorsal and the ventral streams [Perry et al., 2014].
>
> b) Figure 3a. We appreciate your suggestion and have updated the figure with a new colormap.
>
> c) Single-stream with dual tasks. The reviewer raises an interesting question.The brain’s parallel visual streams are divergent anatomical pathways, signifying two separate branches of the visual system. A model that attempts to simulate the human visual system should possess two branches or models, each simulating one stream. Our study’s core objective is not only to explain both streams but also to shed light on the factors differentiating them. Even if a single model trained on two tasks might explain the overall functionality of both streams, it would not be able to delineate their anatomical or functional segregation, a primary focus of our research.
>
> d) Alternative dual-stream models. Thanks for this thoughtful suggestion. We have added an additional alternative model. This alternative model does not explain the ventral and dorsal streams as distinctly as our proposed model, reinforcing our initial conclusion. See Figure R2 attached to the global response.
>
> e) Details about training. Thanks for the suggestion. We refer the reviewer to the global response section, where we provide further details about model training.
>
> f) Saliency prediction. Figure 2 shows results with validation images held out from training. The quantitative measures of the test set of SALICON are AUC: 0.699, SAUC: 0.618. Further qualitative results are presented in Appendix Figure S3.
>
> g) Eye movement data. Unfortunately, the eye movement data is unavailable in the movie fMRI dataset. This prevents us from evaluating a simultaneous prediction of eye movement alongside brain responses. We concur that this would, otherwise, be an interesting analysis.
>
> ***
> i) Indeed, section 4.1 is intended to confirm that training works as intended.
>
> ii) In line 99-100, our intent is to highlight that our model allows “active exploration” via eye movement, since the model includes a branch to control where the look next, distinguishing it from prior works.
>
> iii) We have revised Figure 4 and its caption as suggested.
>
> iv) Our statement of concern is a minor point. A deeper model tends to better explain neural activity [Yamins et al., 2016]. Although our model has fewer parameters, it explains brain responses better than or comparable as deeper models. In case the reviewer has strong objections, we can remove this statement, which would not compromise the main conclusion.

---

> > ### Comment · Reviewer_7ncd · 2023-08-14
> > **Response to author rebuttal**
> >
> > Thank you for addressing my concerns and clarifying various aspects of the paper. I think that the new panels of Figure 4 are a nice addition to the paper, and really seem to demonstrate that for this dataset that main thing that matters is the task and not the input representation. I've updated my score accordingly. Two minor points below (neither are influencing my score, but just wanted to respond to these for your consideration in future drafts).
> >
> > Re: Colormap
> > While I appreciate the attempt to change the color map, there are issues with the non-uniformity and lack of color-blind accessibility. You might want to check out the "uniform diverging pallets" here https://seaborn.pydata.org/tutorial/color_palettes.html and overall discussion on the page about choosing color maps. See also https://colororacle.org/ for a plugin to test whether the map is color-blind friendly.
> >
> > Re: Depth
> > The authors may want to look at more recent work along these lines stating that there is not a direct correlation of neural predictivity with architecture depth (for instance https://www.biorxiv.org/content/10.1101/408385v1.full.pdf). This is where my question came from, as I don't know if there is something specific for dorsal predictions where it is still believed that depth = better predictivity.

---

> > > ### Author Response · Authors · 2023-08-15
> > > **Rebuttal by Authors**
> > >
> > > We are glad that you recognize our responses to the review comments and decide to raise the score. We also value your further comments and will incorporate them into the final version of the paper, should it be accepted. We also welcome any additional feedback or criticism.
> > >
> > > See below for our responses to your minor points.
> > > ***
> > > __Colormap__: Thanks for raising the issues on the colormap. While we should thrive to choose the colormap to represent the results in an unbiased manner, we should also consider accessibility to readers potentially with color vision deficiencies. Thank you for referring us to the related resources. We will consider these resources in making the color figures in the final version of the paper, thriving to balance the scientific accuracy and broad accessibility. Again, thank you.
> > >
> > > __Depth__: Thanks for clarification. We concur with the reviewer’s point that deeper models do not necessarily better predict brain responses. It is not our intention to imply that depth is a reliable factor in determining neural predictivity. Quite the contrary. Results in our current paper, as well as the work by others (Kubilius et al., 2018), demonstrate that model architectures and learning objectives are important factors in explaining brain and behavioral responses, and arguably more important than the depth or scale of the model. However, this is not a key point of our paper. In the final version, we will ensure that our statement accurately reflects the key point with clarity to avoid misinterpretation.

---

### Official Review · Reviewer_ZoBi · 2023-07-07

**Soundness:** 3 good
**Presentation:** 3 good
**Contribution:** 3 good
**Rating:** 5
**Confidence:** 4

**Summary:**

The author introduces a dual-path parallel neural network designed to simulate the hypothesized functions of the ventral and dorsal streams in the human visual system. This network consists of a WhereCNN, which has a wide view to learn spatial attention and direct gaze, and a WhatCNN, which has a local view to learn object recognition. The network is trained using two learning objectives: the WhereCNN is trained with human gaze attention data, while the WhatCNN is trained for object recognition based on local views. The study reveals that the WhereCNN demonstrates better prediction of the dorsal stream voxels, while the WhatCNN performs better in predicting the ventral stream voxels.

**Strengths:**

The alignment of the WhereCNN and WhatCNN with the dorsal and ventral streams provides promising evidence supporting their assumed functions.

**Weaknesses:**

Although the individual networks (WhereCNN and WhatCNN) have been previously proposed and explored for the dorsal and ventral streams, this study is the first to combine them. The paper's assumption of the dorsal stream function may be narrow, as the dorsal stream is known to process spatial information, motion, and action rather than solely focusing on spatial attention and eye fixation decisions.

Given the specific training methodology used for the WhereCNN and WhatCNN, it is not particularly surprising that they show better prediction results for the dorsal and ventral streams, as suggested by earlier studies. Therefore, the results, while valuable in establishing them within a coherent and integrated system, may not be particularly surprising.


**Questions:**

What new insights that this work provides for our understanding of the visual system?
The ideas are not exactly novel, though it is nicely put together, and the alignment with fMRI data is encouraging.
At the end of the day, what does this work teach us about the visual system that we don't already know -- good is supposed to illuminate and elucidate something we don't fully understand.


**Limitations:**

The paper discusses the limitations of the model and highlights areas for future research.

---

> ### Author Rebuttal · Authors · 2023-08-10
>
> Thanks for your summary and constructive feedback.
>
> ***
> 1\. __Novelty__. While we concur that the ventral and dorsal visual streams have been individually studied before, our work is novel in its integrative approach. We model and link the ventral and dorsal streams as two integral parts of a whole system and incorporate a retina-like model to account for the different inputs to the two streams. Our approach is inspired by the neuroscience knowledge that parallel visual processing begins at the retina and progresses through cortical pathways. This holistic view uncovers how retinal inputs (both global and local) alongside behavioral outputs (perception and attention) influence the internal computations and representations emerging from neural processes. While it has limitations, our model is a valuable starting point for modeling parallel visual processing, aiming to deepen our understanding of the functional segregation and integration of the human visual system.
>
> ***
> 2\. __Assumptions on the dorsal stream.__ We recognize the various roles attributed to the dorsal stream, including spatial attention, eye movement, motion processing, recognition, action and more. Despite this diversity, the core functionalities of the dorsal stream that shape its internal representations in distinction from the ventral stream remain a topic of debate. Historically, the understanding of the dorsal stream has shifted between the “where” pathway and the “how” pathway. With this study, we propose that spatial attention and eye movement might represent the core functionalities of the dorsal stream due to their evident correlations with human visual behavior and the relevance to both spatial analysis (“where”) and action-oriented tasks (“how”). Our focus on spatial attention and eye movement is also motivated by the dataset we use - free viewing of natural videos. Given this dataset, evaluating other potential functions would be cumbersome. However, our study does not downplay the potential relevance of other plausible dorsal-stream functions. In future studies, we plan to delve deeper into other functional aspects towards a holistic understanding of the dorsal stream and its interaction with the ventral stream.
>
> ***
> 3\. __Significance.__ Our primary aim is not necessarily the proposition of a groundbreaking theory in neuroscience. Our aim is to demonstrate that a novel integration of interacting models of the eyes, dorsal and ventral streams can explain brain activity during dynamic naturalistic vision at various levels from cortical voxels to regions and streams, extending prior studies which often evaluate different regions in isolation using simple visual tasks. Uniquely, our models are brain inspired and hypothesis-driven, bridging the gap between retinal inputs and behavioral outputs through parallel and hierarchical neural information processing. Our findings both corroborate and falsify existing hypotheses. Notably, we have shown that the distinct retinal inputs to the dorsal and ventral pathways are not sufficient to differentiate their responses during movie viewing. Instead, the distinct learning objectives - namely perception vs. attention - are more significant factors for the functional segregation between these streams. Although our findings may affirm some prior knowledge while challenging others, they are not necessarily predictable from existing literature. Given our model’s adaptability to diverse architectures and objectives, we are confident of its potential to catalyze future research using parallel neural networks.
>
> In terms of modeling the dorsal visual stream using deep neural networks, four prior studies stand out as most relevant to our study. Specifically, the studies by [1,2] focus on training a deep neural network for motion detection or action recognition, while the research by [3] emphasizes predicting self-motion in a 3D environment. The work by [4] demonstrates that employing predictive learning with two distinct streams can lead to representations reminiscent of the dorsal and ventral streams. Distinct from these and other prior studies, to the best of our knowledge, our study is unique for its focus on overt attention as a core functionality of the dorsal stream.
>
> ***
> __References__
>
> [1] Reuben Rideaux and Andrew E Welchman. But still it moves: static image statistics underlie how we see motion. Journal of Neuroscience, 40(12):2538–2552, 2020.
>
> [2] Umut Güçlü and Marcel AJ van Gerven. Increasingly complex representations of natural movies across the dorsal stream are shared between subjects. NeuroImage, 145:329–336, 2017.
>
> [3] Patrick Mineault, Shahab Bakhtiari, Blake Richards, and Christopher Pack. Your head is there to move you around: Goal-driven models of the primate dorsal pathway. Advances in Neural Information Processing Systems, 34:28757–28771, 2021.
>
> [4] Shahab Bakhtiari, Patrick Mineault, Timothy Lillicrap, Christopher Pack, and Blake Richards. The functional specialization of visual cortex emerges from training parallel pathways with self-supervised predictive learning. Advances in Neural Information Processing Systems, 34:25164–25178, 2021.

---

> > ### Comment · Reviewer_ZoBi · 2023-08-20
> > **Thanks for the responses**
> >
> > Thank you for the articulated answers and the citations.  I do recall that there is literature, for example Gustav Deco's work, arguing dorsal stream is for mediating spatial attention and ventral stream is for mediating object attention.  It is still quite a surprise conjecture that dorsal stream but not ventral stream is involved in the controlled of the eye movement, as both streams feedback to V1 as well as to SC for directing eye movements.

---

> > > ### Author Response · Authors · 2023-08-21
> > > **Thank you for the comment.**
> > >
> > > Thank you for sharing additional insight. Indeed, we acknowledge that there are different types of attention (e.g., spatial attention and object attention) likely implemented through different circuit mechanisms. Some of these aspects have been well articulated by Gustavo Deco and Tai Sing Lee as well as others; we will cite related work. In the current modeling work, we focus on spatial attention mediated through the dorsal stream and use spatial attention to drive eye movements. In future studies, we will extend the model by incorporating potential mechanisms of object attention and modeling their roles in gaze behaviors.

---

### Author Rebuttal · Authors · 2023-08-10

We appreciate the time, effort and feedback from all reviewers. We are glad that the review comments are overall positive and seem to merit acceptance. We have also carefully addressed all concerns from the reviewers, by providing new results (shown in the pdf file), discussions, and clarifications, and have performed new experiments according to the suggestions. While we intend to focus mainly on computational neuroscience with brain-inspired neural networks, our model described in this paper would potentially advance the capabilities of adaptive, efficient, and robust computer vision systems – a premise that merits future investigations.

***
0. Correction on Figure 5.

Firstly, we must acknowledge an error in the labeling of Figure 5. The left panel was originally labeled as "WhereCNN” and “Dorsal ROIs" and the right panel as "WhatCNN” and “Ventral ROIs". However, the labels should be reversed. The correct labeling should have the left panel presenting "WhatCNN” and “Ventral ROIs" and the right panel "WhereCNN” and “Dorsal ROIs". We have attached the corrected figure (Figure R3) in the global response section of the supplementary material. This error does not influence our interpretations or the conclusions drawn from the data. We deeply regret this oversight.

***
1. Novelty and Significance

We would like to emphasize the novelty and significance of our work. The goal of our study is to build a better computational model that explains and differentiates the ventral and dorsal streams in the human brain. Compared to numerous studies of the ventral stream, only a few prior works have studied the dorsal stream using neural networks. Further fewer works tried to model both streams altogether. Specifically, in this study we used a fully computable deep neural network to simultaneously model overt visual attention (eye movements) through the dorsal stream, as well as the object recognition by the ventral stream. The combination of the overt attention and the object recognition tasks is also useful for the image computable models in the field of computer vision. Such models can concentrate its resources to informative regions in images or videos, with potential advantages in efficiency and adaptability. Based on the idea, we built an image computable model by capturing key biologically-inspired characteristics of the human visual system (two-stream architecture, retinal transformation and eye movements), and trained each stream with the distinct inputs and learning objectives to mimic their functionalities and behaviors.

Our study reveals that the learned representations from the WhereCNN and WhatCNN correspond to the dorsal and the ventral streams of the human visual systems. We then specifically investigated the key factor responsible for representational differentiation in the two streams, which has been rarely studied with neural network models. Through a set of control experiments to disassociate different factors in the two streams, we report that the major distinction between WhereCNN and WhatCNN representations is driven by the roles that the two streams perform, rather than the distinct retinal inputs they receive. This finding is novel.

In summary, the primary achievements of our research include:
- The construction of a model encapsulating both streams of the human visual system.
- Novelty in modeling the dorsal stream with overt visual attention.
- Novel insights into the distinction in visual stream representations, emphasizing the role of their distinct functionality over diverse inputs.

***
2. Training Details

We thank reviewers' comments regarding the specifics of our training stages. We will include more training details in Appendix Section B in the final version to ensure reproducibility. We have also fully addressed all questions from the reviewers on methodological details or ambiguities. We will the final paper and its appendix reflects full clarity.

Regarding the questions on how stage 3 training was conducted, the brief answer is that we incorporated both object recognition and saliency prediction losses simultaneously, thus it requires a training dataset with labels for both tasks. Because SALICON has labels for both tasks, we used SALICON for stage3. We will ensure greater clarity on this in both the main text and Appendix Section B.

Lastly, we acknowledge the need to clarify the statement in line 165: "After this stage, WhereCNN learns spatial attention from humans.", because it may raise confusions to readers. Here, our intention was to convey that post stage1 training, the WhereCNN is primed to generate saliency maps. There isn't an additional training phase implied. This will be made more explicit in our revised text.

---

### Decision · Program_Chairs · 2023-09-21

**Decision:**

Accept (poster)

**Comment:**

The paper introduces a dual-path architecture to simulate the function of the dorsal and ventral visual streams in the brain, with the WhereCNN guided by spatial attention matching brain activity in the dorsal stream and the WhatCNN guided by object recognition as learning objectives matches responses in the ventral stream better. The reviewers appreciate the overall question and approach, but also raise some issues related to novelty of the work and whether all claims are fully supported. After the response and discussions, some of the reviewers remain somewhat reserved, but all of them see the value of the paper suggest acceptance despite the concerns.